# Meal-Monitoring Systems Using Weight and Temperature Sensors for Elder Residents in Long-Term Care Facilities

**DOI:** 10.3390/ijerph19020808

**Published:** 2022-01-12

**Authors:** Yu Hu, Ji-Eun Joo, Eunju Choi, Leeho Yoo, Dukyoo Jung, Juh-Hyun Shin, Jeong-Ho Kim, Sung-Min Park

**Affiliations:** 1Department of Electronic and Electrical Engineering, Ewha Womans University, Seoul 03760, Korea; huyuu926@gmail.com (Y.H.); wxop01@naver.com (J.-E.J.); jho@ewha.ac.kr (J.-H.K.); 2Graduate Program in Smart Factory, Ewha Womans University, Seoul 03760, Korea; 3College of Nursing, Ewha Womans University, Seoul 03760, Korea; celestial_@naver.com (E.C.); haha_riho@naver.com (L.Y.); dyjung@ewha.ac.kr (D.J.); juhshin@ewha.ac.kr (J.-H.S.)

**Keywords:** dementia, elder-care, LTCF, monitoring system, temperature sensor, weight sensor

## Abstract

This paper presents a few meal-monitoring systems for elder residents (especially patients) in LTCFs by using electronic weight and temperature sensors. These monitoring systems enable to convey the information of the amount of meal taken by the patients in real-time via wireless communication networks onto the mobile phones of their nurses in charge or families. Thereby, the nurses can easily spot the most patients who need immediate assistance, while the families can have relief in seeing the crucial information for the well-being of their parents at least three times a day. Meanwhile, the patients tend to suffer burns of their tongues because they can hardly recognize the temperature of hot meals served. This situation can be avoided by utilizing the meal temperature-monitoring system, which displays an alarm to the patients when the meal temperature is above the reference. These meal-monitoring systems can be easily implemented by utilizing low-cost sensor chips and Arduino NANO boards so that elder-care hospitals and nursing homes can afford to exploit them with no additional cost. Hence, we believe that the proposed monitoring systems would be a potential solution to provide a great help and relief for the professional nurses working in elder-care hospitals and nursing homes.

## 1. Introduction

In 2000, the global population aged over 60 was only 11%, and it was predicted that it would be doubled by 2050, reaching 22% of the total population. Accordingly, the forecast that the need of long-term care facilities (LTCFs) would surge has become a reality [1]. In 2017, South Korea entered into an aging society by the rapid proliferation of the aged population. This fact was recorded as the fastest among all the Organization for Economic Cooperation and Development (OECD) member countries. Now, it seems that South Korea might become a super-aging society in 2025 such that the population aged over 60 would represent more than 20% of the population [2].

This phenomenon continually caused the rapid increase of the elders admitted to LTCFs, i.e., 24,195 elders in 2005 versus 174,015 elders in 2020 [3]. Particularly, 75~88% of the elders in LTCFs suffer from cognitive impairment and are affected by diseases such as dementia, stroke, and Parkinson’s disease [4,5]. About 56% of the elders in LTCFs have experienced gradual decrease in physical function during the period of two years after admission [6,7]. Even after three months of admission, their abilities of daily living have been shown to degrade when compared to their initial evaluation [8]. This decrease of physical function naturally leads to incontinence and nutrition problems [9]. As an example, 50~79.7% of the elders in LTCFs can suffer from urinary incontinence [10,11], and 10.5% of the non-malnourished might become malnourished within six months [12].

Besides, the elders in LTCFs may be at high risk of chronic dehydration due to their low capability of fluid intake, which is commonly observed [13].

Nutrition has been generally recognized as an important nursing issue for elders [14]. Particularly, it is extremely crucial for the elders in LTCFs because more than two-thirds suffer from dementia, which therefore leads to the common issues of weight loss and malnutrition [15]. As reported in Rullier et al. (2013) [16], 58.9% of the elders with dementia staying in LTCFs show poor nutritional status. However, public elder-care hospitals or nursing homes may be short of professional manpower, such as registered nurses. As of 2018, the number of nurses in elderly care facilities in South Korea was 1472, which indicates that there is one registered nurse available for care per every 111 elders [17]. In these inadequate situations, we believe that it would be a great help and relief to nurses to provide a meal-monitoring system that enables them to easily figure out how much food each patient has been eating in real-time at least three times a day. It would also help nurses spot the patients who take meals inadequately and hence optimize their capabilities to nurse patients closely [18].

Meanwhile, many elders in LTCFs undergo significant changes of sensory function, thereby resulting in substantial weight loss [19]. As a potential example, high risk of oral problems might occur due to the ingestion of hot food in situations when either nurses or caregivers cannot closely observe the patients. It is well known that maintaining oral hygiene is imperative for food intake. Otherwise, it gives rise to nutritional problems and deteriorates overall health, potentially becoming fatal [20].

Previously, smart-phone-based food intake-monitoring systems were introduced to measure chewing or biting patterns and speed for obese people and elderly people [21,22]. A wearable device was presented to measure how elderly people eat, chew, and swallow food [23]. Additionally, a real-time based system was realized to measure patterns and quantity of food intake for Alzheimer-type dementia patients [24]. In addition, qualitative research, instead of a monitoring technology system, was suggested to emphasize the nurse staffing factor on food intake for dementia residents in nursing homes [25].

In this paper, we propose two meal-monitoring systems using electronic sensor devices in order to implement the automatic functions of both alerting the patients the meal temperature and conveying the food diary information of the patients (or the elders in LTCFs) to their nurses and/or families via wireless transfer in real time [26,27]. First, a meal weight-monitoring system was realized with an Arduino NANO board and a weight sensor that can figure out how much food is taken in every five seconds until the patients finish their meal. The size of the meal weight-monitoring system can be designed to fit a normal food-tray, while its height should be as thin as possible for users. Various tests of these meal weight-monitoring systems were conducted in a simulation ward for the purpose of future practical usage. Furthermore, a non-contact optical sensor was utilized with the Arduino NANO board to obtain the meal temperature. Due to the limited range of the exploited non-contact optical sensor and also for more precise measurements, the meal temperature-monitoring system is located in proximity to the hot meal.

Section 2 describes the details of the proposed meal-monitoring systems. First, a meal-weight monitor is fully described. Then, a meal-temperature monitor is explained. Section 3 demonstrates the measured results of these two meal-monitoring systems together with the test results of the combined meal-monitoring system on a clinical bed. Finally, a conclusion follows.

## 2. Meal Monitoring Systems

In this section, two types of meal-monitoring systems are presented, which include a meal weight-monitoring system and a meal temperature-monitoring system.

### 2.1. Meal Weight Monitoring System

The proposed meal weight-monitoring system consists of four load-cells, a front-end analog amplifier with an analog-to-digital converter (ADC), and an Arduino NANO board with Bluetooth module. Figure 1 shows the picture of a load-cell (which is limited to the maximum weight of 50 kg) and its equivalent circuit, in which it is clearly seen that a weight can change the variable resistance of the load-cell, thus generating a different output voltage from a reference voltage.

This voltage output is sensed by a following analog amplifier with an ADC (HX711 in this work). Then, an Arduino NANO board is connected to measure the value of weight and transfer the data wirelessly over to a Bluetooth-enabled smart device every 5 s.

Now, the weight of a meal tray for each elder patient is measured by utilizing four load-cells. These load-cells should be located at the corners of the tray, and therefore, the corresponding four output voltages can be simultaneously detected by the ADC. Then, the following Arduino NANO board generates the final weight onto a monitor that can be either a small display or a monitor. In addition, a Bluetooth module compatible with an Arduino board can be utilized to transmit the weight information onto the mobile phones of both nurses and families (as shown in Figure 2) so that they can figure out the amount of food that each patient is taking in during the meal time [26]. This would certainly be a great relief to the families and also a crucial information to nurses in charge.

Yet, this configuration may require a critical process called “initial system calibration”, which sets a reference weight for precise measurements. Namely, a calibration factor should be tuned for proper weight information. In this work, the calibration factor was set to 104,000 in the Arduino code. This calibration process typically resets the initial weight to 0 g for start and then measures the precise weight of a meal tray. Here, it should be noted that the weight data can be sensitive to the location of four load-cells under a meal tray, and therefore, each load-cell should be attached to each corner of the tray. Certainly, a tray-shape scale with an identical size would be preferred for this type of weight measurement.

Nonetheless, there exists a merit in this proposed meal weight-monitoring system, which is that the relative amount of food left on a tray can be measured even if the reference weight cannot be calibrated precisely. In other words, even in the cases that nurses forget to tune the calibration factor for precise measurements by any reason, our monitoring system can generate the difference between the original meal weight and the final weight so that the nurses in charge can easily figure out the amount that each patient has eaten. Figure 3 shows an example with an uncorrected calibration factor, where the initial weight is read as −3.5 kg, while the weight of an empty tray when finished is recorded as −2.0 kg. Although the absolute value of meal weight taken by the patient is not shown, nurses and families can see clearly whether the patient has eaten or not.

### 2.2. Meal Temperature Monitoring System

As aforementioned, the temperature of a meal tray should be measured to protect the patient’s tongue and maintain oral hygiene because most dementia patients can hardly recognize how hot their meals are. Research evidence indicates that older adults can experience declines in thermal sensitivity, which may prevent them from adequately judging how hot food and drinks can be, and therefore, this may lead to oral thermal injuries. Especially, the authors’ experience in nursing homes or in nursing reveals that patients swallow hot soups with no hesitation, and their tongues are frequently damaged. Moreover, most elderly patients tend to stop eating when the meals get cold.

Hence, we developed a meal temperature-monitoring system using a bipolar junction transistor (BJT) sensor and an Arduino NANO board so that the elderly patients can be alerted to hot temperatures, and the nurses can also recognize the necessity of rewarming cold food for some patients. In particular, the alert message can be displayed either on-screen or heard through the hearing aid of the patients.

Figure 4 shows a conceptual architecture of the proposed meal temperature-monitoring system in which a non-contact optical sensor and an LCD display are connected to an Arduino NANO board. Besides, a Bluetooth module compatible with an Arduino board can be utilized to transmit the temperature information onto the mobile phones of nurses in charge so that the nurses can serve the patients more effectively, e.g., the cold food can be rewarmed in a timely manner for elderly patients.

In this work, an optical sensor was exploited to emit infrared lights to targets for temperature measurements in the range between −70 °C and 380 °C with the accuracy of 0.5 °C. The reference temperature was set to 30 °C. Above this reference temperature, any form of alarm can be displayed on the LCD screen. Additionally, another was set to 20 °C, below which it sends an alarm to nurses for food replacement. 

## 3. Measured Results

In this section, the practical implementation of the proposed meal-monitoring systems is presented along with their measured results.

### 3.1. Meal Weight-Monitoring System

Figure 5 shows an implementation of a weight-monitoring module with a single load-cell along with its measured results obtained by pressing a single load-cell with a human hand, where it is clearly seen that the variation of weight is recorded.

Figure 6 demonstrates a practical implementation of the proposed meal weight-monitoring system, where four load-cells are located under the corners of a meal tray. Then, it demonstrates the real-time measurements (after calibration) of the meal weight left on a food tray: (a) the initial weight of a meal tray with food, (b) a meal weight in the middle of meal intake, and (c) the final meal weight when finished.

In Figure 6a, the initial weight of 1.8 kg is recorded. After a few minutes, Figure 6b records 1.6 kg (which is 200 g reduced from the initial weight) and shows the percentage of 11%, indicating that the designated patient has eaten 11% of the provided meal. Finally, Figure 6c presents the final weight of an empty tray, i.e., 1.0 kg, which is shared with nurses and families. It provides the relative percentage of 43%, which is the ratio of 1.0 kg to the initial weight of 1.8 kg.

Hence, nurses and families can see clearly in real time how much food the patient has eaten. In addition, all the acquired information can be continuously transferred wirelessly to the mobile phones of nurses in charge and the patient’s family. The left picture in Figure 6d shows the screen of a mobile phone with the information of the weight variation through the process of meal intake. The relative difference of weight is also clearly shown. The right picture in Figure 6d corresponds to the case of Figure 6c.

### 3.2. Meal Temperature-Monitoring System

In order to demonstrate an example of temperature measurements, we exploited a non-contact optical sensor and an Arduino NANO board with an LCD display. Here, the reference temperature was set to 30 °C. Figure 7 demonstrates the measured results of the meal temperature-monitoring system for three different temperatures. First, Figure 7a shows the temperature of 40 °C, which is high above the reference. Therefore, it shows a form of alarm (XX mark in this work) on an LCD screen for hot temperature. Second, Figure 7b demonstrates the mild temperature of 20 °C, with no alarm turned on yet, rather showing a heart sign in this work. Third, when a meal temperature below 20 °C is detected, the monitoring system sends an alarm signal to nurses so that the cold meal can be replaced with a hot one if necessary. Third, Figure 7c depicts the temperature of 17 °C, which is below the other reference of 20 °C. Then, an X mark is displayed on the LCD screen as a token of alarm. The size of an LCD display utilized in this work can be replaced with a much larger one for senile dementia patients in order for them to recognize the alarm vividly.

Figure 8 presents the wireless transfer of the temperature information onto the mobile phone via a Bluetooth network so that nurses can easily be alerted to the need for replacing a cold meal with a hot one. Furthermore, families can see the state of food temperature served in elder-care hospitals or nursing homes in real time.

### 3.3. Combined Meal-Monitoring System

The aforementioned two types of monitoring systems were combined and placed on a clinical bed in a simulation ward for more practical testing, as shown in Figure 9. This combined monitor system concurrently measures the mean temperature and the amount of meal intake in real time. Here, we set the reference temperature to 35 °C, below which senile dementia patients might detect soup as cold and demand it to be replaced. Figure 9a is the case of before-the-meal intake, demonstrating on a smart phone display that the hot meal temperature is measured to be 46.85 °C, while the meal weight variation is 0%. Figure 9b is the case of after the meal, exhibiting the cold temperature of the empty tray and confirming 100% meal intake of the patient.

## 4. Conclusions

We implemented two meal-monitoring systems for elderly patients residing in LTCFs by utilizing electronic sensors together with an Arduino NANO board in order to measure the meal weight left on a food tray in real time when the patients are eating and also to measure the temperature of a hot meal. Thereby, the amount of food taken in by the patients is easily recorded and even transferred wirelessly via either Bluetooth or Wi-Fi networks onto the mobile phones of both the nurses in charge and the families of the patients instantly. Moreover, the hot temperature of the served meal is displayed on screen so that the patients can be alerted to avoid the burning of their tongues and to maintain their oral hygiene. However, it may be feasible that those who are cognitively impaired may ignore this warning. Therefore, a sound-warning system can be additionally implemented to aid their recognition, which requires future studies. Finally, we combined these two types of monitoring systems into one and placed it on a clinical bed for more practical testing, which confirms the efficient monitoring of meal weight and temperature. Hence, it can be concluded that the proposed meal-monitoring systems will help elderly patients take in more food and thus lengthen their precious lives and benefit their health. Furthermore, they can provide a great help and relief not only for the families of the patients but also for nurses in elder-care hospitals and nursing homes.

## Figures and Tables

**Figure 1 ijerph-19-00808-f001:**
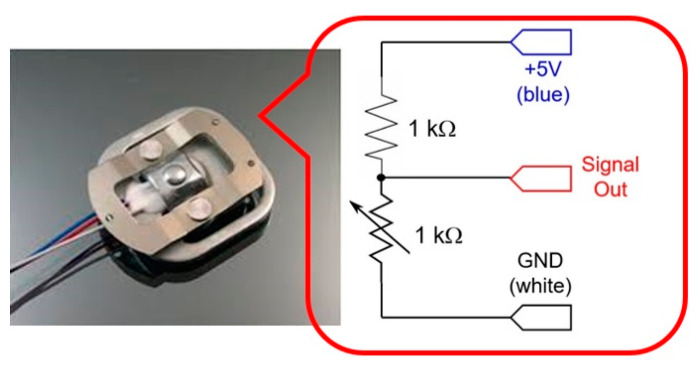
Picture of a load-cell and its equivalent circuit.

**Figure 2 ijerph-19-00808-f002:**
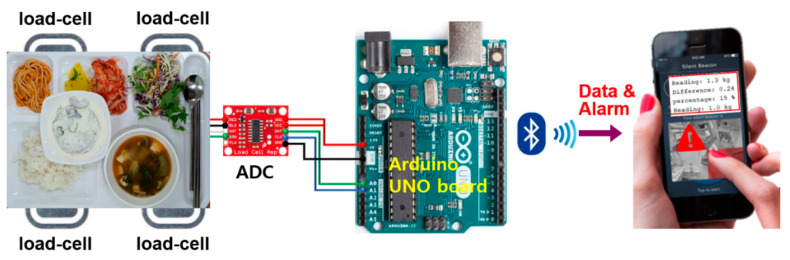
Conceptual architecture of a meal weight monitoring system.

**Figure 3 ijerph-19-00808-f003:**
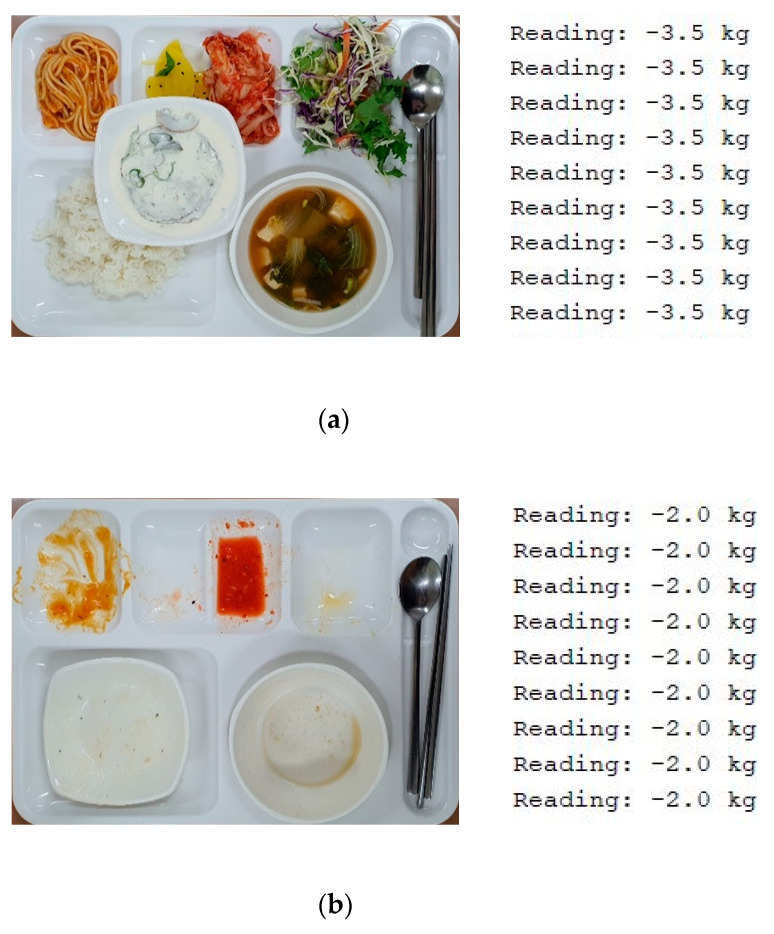
Meal weight measurements with no calibration factor: (**a**) initial meal and (**b**) after the meal time.

**Figure 4 ijerph-19-00808-f004:**
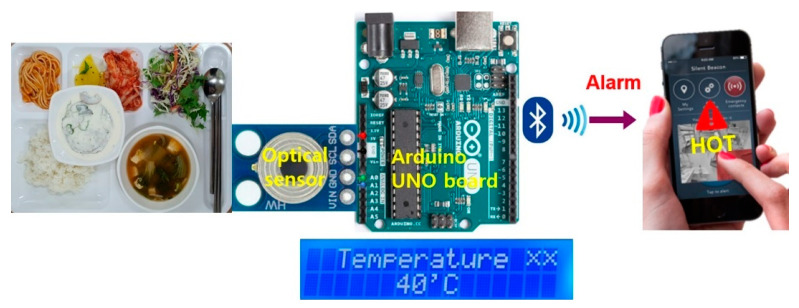
Conceptual architecture of a meal temperature-monitoring system.

**Figure 5 ijerph-19-00808-f005:**
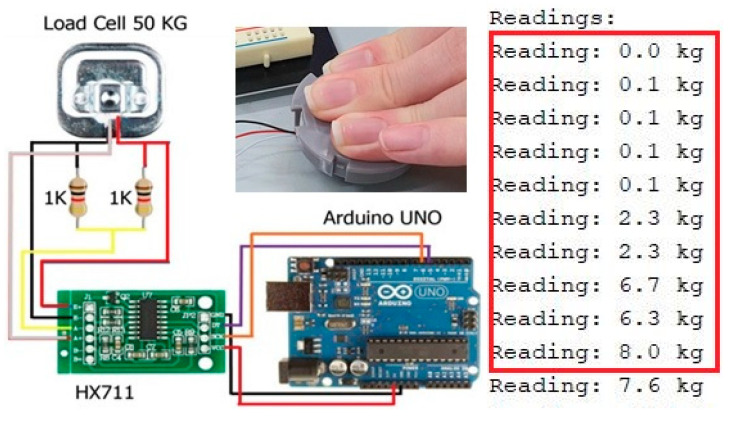
Example of a weight-measuring module with a single load-cell.

**Figure 6 ijerph-19-00808-f006:**
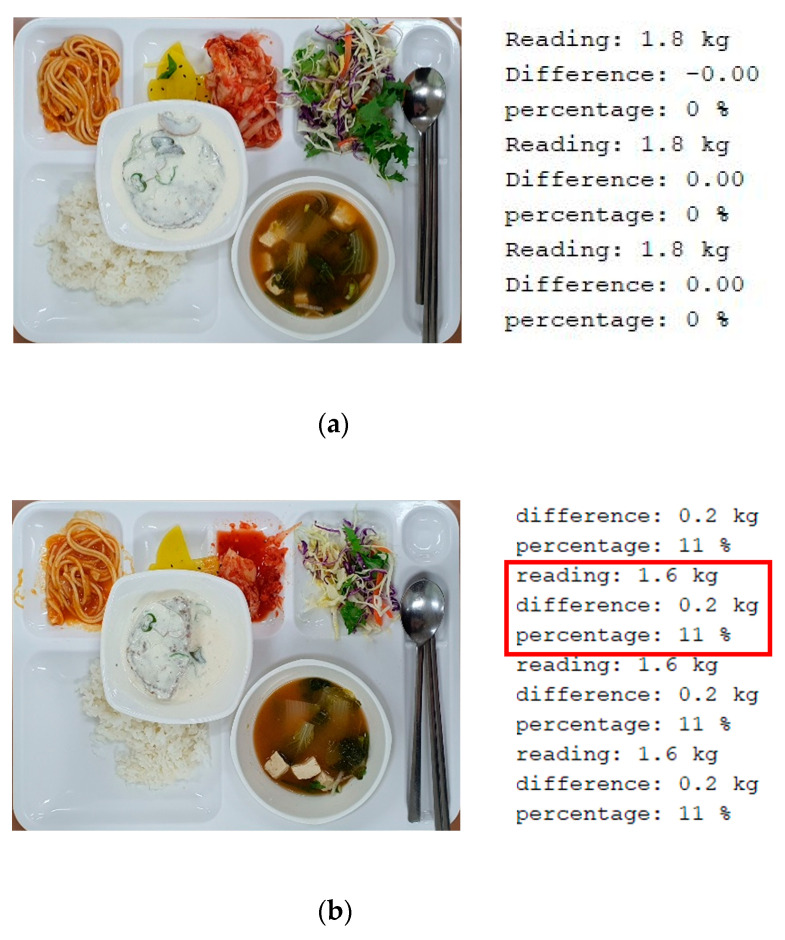
Meal-weight measurements with a corrected calibration factor: (**a**) initial meal, (**b**) meal in the middle, (**c**) after the meal time, and (**d**) wireless transfer via Bluetooth.

**Figure 7 ijerph-19-00808-f007:**
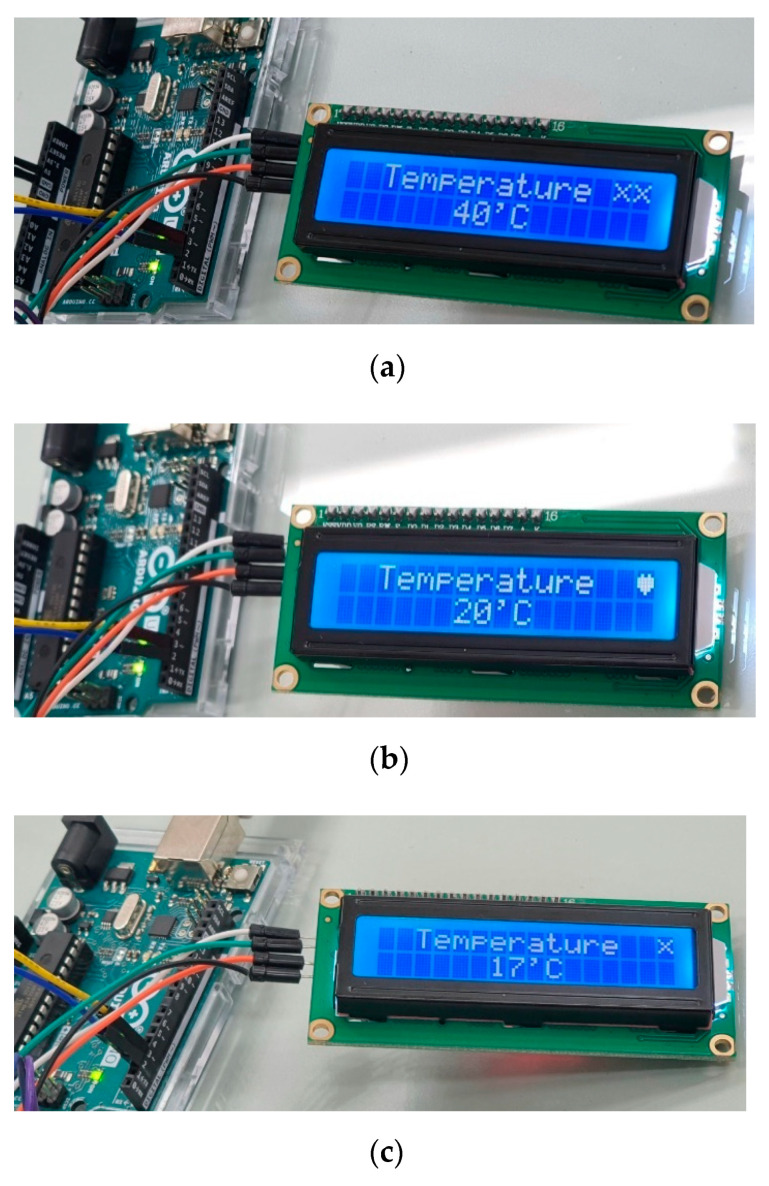
Meal temperature measurements by using an LCD display: (**a**) above (XX), (**b**) normal (heart), and (**c**) below (X) the reference temperature.

**Figure 8 ijerph-19-00808-f008:**
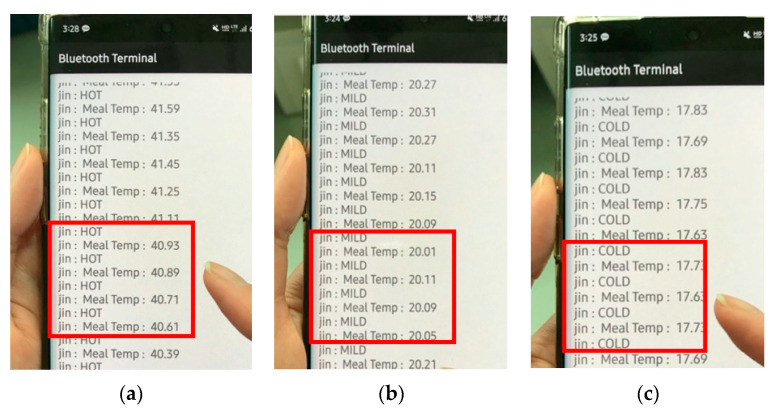
Wireless transfer over a mobile phone by using Bluetooth network: (**a**) hot (over 40 °C), (**b**) mild (in between), and (**c**) cold (below 20 °C) temperatures.

**Figure 9 ijerph-19-00808-f009:**
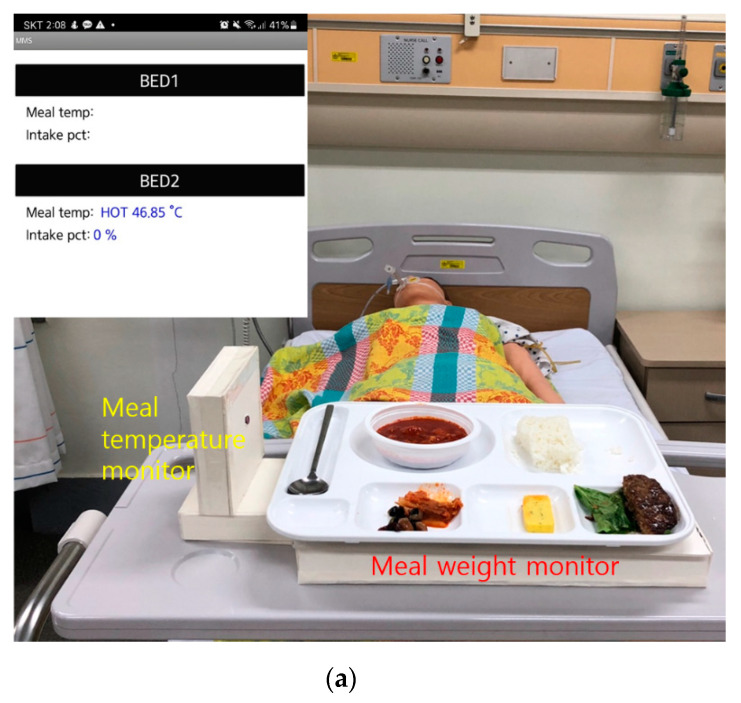
Combined meal monitoring system on a clinical bed in a simulation ward: (**a**) before and (**b**) after the meal intake of a patient.

## Data Availability

Not applicable.

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
