# Peer review of "Meal-Monitoring Systems Using Weight and Temperature Sensors for Elder Residents in Long-Term Care Facilities"

_ijerph, 2022, doi:10.3390/ijerph19020808_

Round 1
Reviewer 1 Report
In this study the authors proposed a meal monitoring system using electronic weight and temperature sensors, which may provide a promising solution to help nurses in elder-care hospitals and nursing homes in monitoring the elderly’s diet intake and preventing burns. The concept of this system is innovative, however, there are some concerns to be addressed.
- First of all, the current proposed system hasn’t been compared with the standard method with regard to meal temperature. How many points needed for signaling a hot risk? The accuracy? Temperatures for food taken into mouth and food on the plate mean differently.
- Has the system been used in real LTCF circumstances to test its practicability? What about the effects?
- The transmission distance of Bluetooth signal is limited (usual within 10 meters), and this may compromise its application.
- Are there any similar systems reported before? Previous research should be cited?
- The introduction should give a comprehensive description about the techniques being proposed, rather than focus on ageing and the related problems. Figure 1 is not necessary.
- The structure of manuscript should fit for the journal requests.
Author Response
Reviewer: 1
In this study the authors proposed a meal monitoring system using electronic weight and temperature sensors, which may provide a promising solution to help nurses in elder-care hospitals and nursing homes in monitoring the elderly’s diet intake and preventing burns. The concept of this system is innovative, however, there are some concerns to be addressed.
1. First of all, the current proposed system hasn’t been compared with the standard method with regard to meal temperature. How many points needed for signaling a hot risk? The accuracy? Temperatures for food taken into mouth and food on the plate mean differently.
-> (ans.) Thanks a lot for this comment.
(1) The meal temperature monitoring system measures the temperature in every one second, such that a warning can be sent as soon as the measured temperature is above the reference (e.g., 35 oC in Fig. 10). Namely, even one point is above the reference, a warning is transmitted.
(2) In our test, another commercial thermometer was utilized for comparison, in which we have confirmed the accuracy of our meal-temperature monitor. Nonetheless, the accuracy of the utilized optical sensor (MLX90614-DCI) is 0.5 oC, as described in Section 2.2.
(3) As commented, the temperatures of food taken into mouth would be slightly different from those of food on plates. However, most dementia patients have tendency to take hot-meal (e.g., hot soup) with no hesitation, so that they might burn their tongues and damage their oral hygiene. Hence, we authors believe that it cannot be overestimated if a warning for hotter-temperature of ‘food on the plate’ can be instantly delivered to those patients so that they would be able to cope with hot-meals cautiously.
2. Has the system been used in real LTCF circumstances to test its practicability? What about the effects?
-> (ans.) These systems have not yet been utilized in real LTCF circumstances. We just have tested these systems in a simulation ward, and the results are shown in Fig. 10. Nonetheless, we are quite certain that the meal monitoring system would function in real LTCF circumstances as well as anticipated so that the nurses could figure out all the necessary information from their mobile phones.
3. The transmission distance of Bluetooth signal is limited (usual within 10 meters), and this may compromise its application.
-> (ans.) As commented, the transmission distance of Bluetooth network would be limited to 10 meters or less. So, WiFi network of which signal transmission coverage is much wider might be utilized instead. However, in terms of battery power maintenance of sensor devices, WiFi is less efficient due to its longer distance of signal transmission. That’s why we have selected Bluetooth network for sure. Once a registered nurse enters a designated ward, the necessary data of patients would be securely delivered to her/his mobile phone. However, if necessary, WiFi networks could be utilized to relay the data over to distant places.
4. Are there any similar systems reported before? Previous research should be cited?
-> (ans.) Thanks a lot for this comment. We have added and cited previous researches in the revised manuscript as below.
“Previously, smart-phone based food intake monitoring systems were introduced to measure chewing or biting patterns and speed for obese people and elderly people [21, 22]. A wearable device was presented to measure how elderly people eat, chew, and swallow food [23]. A real-time based system was realized to measure patterns and quantity of Alzheimer-type dementia patients [24]. Also, instead of a monitoring technology system, qualitative research was suggested to emphasize the nurse staffing factor on food intake for dementia residents in nursing homes [25].”
5. The introduction should give a comprehensive description about the techniques being proposed, rather than focus on ageing and the related problems. Figure 1 is not necessary.
-> (ans.) As commented, Fig. 1 is now deleted in the revised manuscript. Even though we described the proposed techniques in the original manuscript (lines 70 ~ 76), we have added some more comprehensive description in the revised manuscript as below.
“In this paper, we propose a couple of meal monitoring systems using electronic sensor devices in order to implement the automatic functions of both alerting the patients the meal temperature and conveying the food diary information of the patients (or the older people in LTCFs) to their nurses and/or families via wireless transfer in real-time [26,27]. First, a meal weight monitoring system is realized with a low-cost Arduino NANO board and a weight sensor that can figure out how much food to be taken in every 5 seconds until the patients finish their meal. The size of the meal weight monitoring system can be designed to fit a normal food-tray, while its height should be as thin as possible for users. Various tests of these meal weight monitoring systems are conducted even in a simulation ward for the purpose of future practical usage. Also, a non-contact optical sensor is utilized with the Arduino NANO board to obtain the meal temperature. Due not only to the limited range of the exploited non-contact optical sensor, but also for more precise measurements, the meal temperature monitoring system is located in proximity to hot meal (e.g., as shown in Fig. 10).”
6. The structure of manuscript should fit for the journal requests.
-> (ans.) Thanks a lot for this comment. We have tried our best to fit the structure of the revised manuscript for the journal requests.

Reviewer 2 Report
This was a fascinating early concept study to read, the interdisciplinary nature of the study is extremely interesting. Gerontech is a growing field, and the carer as the end user is a worthwhile consumer group to consider. The authors used existing technology and customised them to suit specific needs identified in aged care. The study was conducted from a predominantly engineering perspective, and would benefit from more nursing and consumer input in later stages. In general, the authors should try to improve the writing quality of the manuscript. I recommend editing with my feedback below in mind. Please note that I did not point out every error (especially minor language issues), I have provided examples.
Intro
Please spell out OECD in full before using acronyms.
Line 37: “…suffer FROM cognitive impairment…”
Iine 40-41: “have been DOCUMENTED TO DEGRADE when compared to…”
Line 43: “might suffer FROM urinary incontinence…”
Suggest changing “might” to “can”. The word “might” has some likelihood connotations, where as you are directly reporting percentages.
Line 53: “…leads to THE COMMON ISSUES of weight loss…”
Line 54, please write Rullier et al (2013) and then place the [16] citation
Line 56-57 “…nursing homes may BE SHORT of professional…”
Line 55-57: is replacement of registered nurses with nursing assistants a major issue in South Korea and elsewhere? The way this scenario has been described does not make it clear that this is a phenomenon happening right now that is resulting in lower registered nurse numbers in these facilities. Please consider rephrasing this sentence to connect more smoothly with the next sentence.
Line 62: suggest to change the word “barely” to something more specific, e.g. “low/inadequate meal intake”.
Line 65: “Especially, high risk of oral problems occurs due to the ingestion of hot food in situations when either nurses or caregivers can- 66 not closely observe the patients.” – are there any empirical evidence to support this? If not, I suggest editing this sentence to serve as an potential example of the sensory problems you have raised.
Line 70: suggest changing “a couple of” to “a number of” or just state the number, this will be a more appropriate tone for an academic paper.
Line 72: “older people” is used instead of “elders”, I suggest the authors use the same term consistently throughout the paper.
Line 74: “low-cost” is relative and not an objective description unless you have conducted cost comparisons, I suggest leaving this out when you write about the equipment throughout the paper.
Section 2:
Figure 3: can you spell out UL/UR/LL/LR fully OR include a figure legend, it’s best to not have acronyms in figures even if they seem simple. I suggest replacing the HX711 with “analog-to-digital converter” in the figure and elsewhere in the paper.
Line 106-107 “…a Arduino NANO boars is CONNECTED TO MEASURE the value of weight and transfer the data WIRELESSLY over to A BLUETOOTH-ENABLED SMART DEVICE EVERY 5 SECONDS.”
Line 130: “…each patient HAS eaten.”
Section 2.2:
Line 138: is there a direct relationship between the temperature of food and oral hygiene? Or do you mean oral health in general? The term “senile” has negative connotations in English, I suggest changing it to “cognitively impaired”.
Lines 139-141: the example with soup is useful, but it’s very specific and it would not be the only situation in which this happens. My feedback for this is the same as the feedback for a similar statement in line 65. This may be anecdotal evidence, but empirical evidence is not as straight forward (refer to https://link.springer.com/article/10.1007/s10266-021-00594-4 and https://onlinelibrary.wiley.com/doi/abs/10.1288/00005537-199202000-00001). I suggest starting by stating the research evidence that older adults CAN experience declines in thermal sensitivity, which MAY prevent them from adequately judging how hot food and drinks can be, and therefore this MAY lead to oral thermal injuries, and then end with a statement about the anecdotal evidence from where you have observed this, e.g. the authors’ experience in a nursing home or as a nurse.
Line 156: please clarify what MLX90614-DCI refers to. Remember your readers won’t necessarily have an engineering background, always use the common name of an instrument instead of a model number/code, e.g. use analog-to-digital converter instead of HX711.
Section 3:
Line 178: 1.0kg should be 0.8kg?
All figures display weight displacements and differences: I suggest cropping the numbers to only one set for easier reading, or use the red square to only highlight one set of readings. Right now you have multiple similar readings all clumped together, which can overwhelm readers and take them longer to understand.
Line 203: the way you have describe the non-contact optical sensor followed by the model number in brackets is what I am suggesting you do for line 156.
Section 4 conclusion:
Line 251: A couple? Do you mean the 2 systems you have described in the study, or that you tried this combined system on a number of residents in the LTCF for a number of meals?
Line 258: one potential problem with the visual warning of temperature is that if the person is significantly impaired in cognition, then they will may also ignore this warning. This may be outside the scope of your study, but it’s worth nothing for follow up studies and implementation studies.
I’d like the authors to clarify whether this study was demonstrating a systems described or if it also involved trialling the system on real people over a number of meals. I was not completely sure based on the way the study was written.
While I understand communication articles are shorter than research article formats, the authors should still provide a brief discussion section to address the strengths and weaknesses of the systems, as well as issues regarding their implementation in real world situations. E.g. you mentioned customising trays with cell placement for weight monitoring, but does that mean LTCFs have to change all their trays? How would that influence costs?
Author Response
Reviewer: 2
This was a fascinating early concept study to read, the interdisciplinary nature of the study is extremely interesting. Gerontech is a growing field, and the carer as the end user is a worthwhile consumer group to consider. The authors used existing technology and customized them to suit specific needs identified in aged care. The study was conducted from a predominantly engineering perspective, and would benefit from more nursing and consumer input in later stages. In general, the authors should try to improve the writing quality of the manuscript. I recommend editing with my feedback below in mind. Please note that I did not point out every error (especially minor language issues), I have provided examples.
Intro
Please spell out OECD in full before using acronyms.
Line 37: “…suffer FROM cognitive impairment…”
Iine 40-41: “have been DOCUMENTED TO DEGRADE when compared to…”
Line 43: “might suffer FROM urinary incontinence…”
Suggest changing “might” to “can”. The word “might” has some likelihood connotations, where as you are directly reporting percentages.
Line 53: “…leads to THE COMMON ISSUES of weight loss…”
Line 54, please write Rullier et al (2013) and then place the [16] citation
Line 56-57 “…nursing homes may BE SHORT of professional…”
-> (ans.) Thanks a lot for this correction. We have corrected all these in the revised manuscript.
Line 55-57: is replacement of registered nurses with nursing assistants a major issue in South Korea and elsewhere? The way this scenario has been described does not make it clear that this is a phenomenon happening right now that is resulting in lower registered nurse numbers in these facilities. Please consider rephrasing this sentence to connect more smoothly with the next sentence.
-> (ans.) Thanks a lot for this comment. For clarity and smooth connection with the next sentence, we have deleted ‘under circumstances where registered nurses can be legally replaced with nursing assistants’.
Line 62: suggest to change the word “barely” to something more specific, e.g. “low/inadequate meal intake”.
-> (ans.) As commented, we have corrected all these in the revised manuscript.
Line 65: “Especially, high risk of oral problems occurs due to the ingestion of hot food in situations when either nurses or caregivers can- 66 not closely observe the patients.” – are there any empirical evidence to support this? If not, I suggest editing this sentence to serve as a potential example of the sensory problems you have raised.
-> (ans.) As commented, we have modified the sentence as below.
“As a potential example, high risk of oral problems might occur due to the ingestion of hot food in situations when either nurses or caregivers cannot closely observe the patients.”
Line 70: suggest changing “a couple of” to “a number of” or just state the number, this will be a more appropriate tone for an academic paper.
Line 72: “older people” is used instead of “elders”, I suggest the authors use the same term consistently throughout the paper.
-> (ans.) As commented, we have corrected all these in the revised manuscript.
Line 74: “low-cost” is relative and not an objective description unless you have conducted cost comparisons, I suggest leaving this out when you write about the equipment throughout the paper.
-> (ans.) As commented, we have deleted ‘low-cost’ throughout the paper.
Section 2:
Figure 3: can you spell out UL/UR/LL/LR fully OR include a figure legend, it’s best to not have acronyms in figures even if they seem simple. I suggest replacing the HX711 with “analog-to-digital converter” in the figure and elsewhere in the paper.
-> (ans.) As commented, we have modified Fig. 3 in the revised manuscript.
Line 106-107 “…a Arduino NANO board is CONNECTED TO MEASURE the value of weight and transfer the data WIRELESSLY over to A BLUETOOTH-ENABLED SMART DEVICE EVERY 5 SECONDS.”
Line 130: “…each patient HAS eaten.”
-> (ans.) As commented, we have corrected all these in the revised manuscript.
Section 2.2:
Line 138: is there a direct relationship between the temperature of food and oral hygiene? Or do you mean oral health in general? The term “senile” has negative connotations in English, I suggest changing it to “cognitively impaired”.
-> (ans.) We meant ‘oral health in general’.
-> As commented, we have replaced ‘senile’ with ‘cognitively impaired’ in the revised manuscript.
Lines 139-141: the example with soup is useful, but it’s very specific and it would not be the only situation in which this happens. My feedback for this is the same as the feedback for a similar statement in line 65. This may be anecdotal evidence, but empirical evidence is not as straight forward (refer to https://link.springer.com/article/10.1007/s10266-021-00594-4 and https://onlinelibrary.wiley.com/doi/abs/10.1288/00005537-199202000-00001). I suggest starting by stating the research evidence that older adults CAN experience declines in thermal sensitivity, which MAY prevent them from adequately judging how hot food and drinks can be, and therefore this MAY lead to oral thermal injuries, and then end with a statement about the anecdotal evidence from where you have observed this, e.g. the authors’ experience in a nursing home or as a nurse.
-> (ans.) As commented, we have modified the sentence as below.
“Research evidence indicates that older adults can experience declines in thermal sensitivity, which may prevent them from adequately judging how hot food and drinks can be, and therefore this may lead to oral thermal injuries. Especially, the authors’ experience in a nursing home or as a nurse reveals that they swallow hot soups with no hesitation, and their tongues are frequently damaged. Also, most elder patients tend to stop eating when the meals are getting cold.”
Line 156: please clarify what MLX90614-DCI refers to. Remember your readers won’t necessarily have an engineering background, always use the common name of an instrument instead of a model number/code, e.g. use analog-to-digital converter instead of HX711.
-> (ans.) As commented, we have modified the sentence as below.
“In this work, an optical sensor is exploited to emit infrared lights to targets for temperature measurements in the range between -70 oC and 380 oC with the accuracy of 0.5 oC.”
Section 3:
Line 178: 1.0kg should be 0.8kg?
All figures display weight displacements and differences: I suggest cropping the numbers to only one set for easier reading, or use the red square to only highlight one set of readings. Right now you have multiple similar readings all clumped together, which can overwhelm readers and take them longer to understand.
-> (ans.) As commented, we have corrected those in Fig. 7 (Fig. 6 in the revised manuscript). The final weight is 1.0 kg. Only the difference is 800 g, which is the amount of meal taken.
Line 203: the way you have describe the non-contact optical sensor followed by the model number in brackets is what I am suggesting you do for line 156.
-> (ans.) As commented, we have deleted ‘the model number’ in the revised manuscript.
Section 4 conclusion:
Line 251: A couple? Do you mean the 2 systems you have described in the study, or that you tried this combined system on a number of residents in the LTCF for a number of meals?
-> (ans.) We meant two monitoring systems because we have initially demonstrated each system and afterwards combined two systems into one. Therefore, we have replaced ‘a couple of’ with ‘two’.
Line 258: one potential problem with the visual warning of temperature is that if the person is significantly impaired in cognition, then they will may also ignore this warning. This may be outside the scope of your study, but it’s worth noting for follow up studies and implementation studies.
-> (ans.) Thanks a lot for this comment. We have added some more about future work as below.
“Also, the hot temperature of the served meal is displayed on screen so that the patients can be alert to avoid the burning of their tongues and to maintain their oral hygiene. Yet, it may be feasible for those who are impaired in cognition to ignore this warning. Therefore, sound-warning system can be additionally implemented to aid their recognition, which would be our future work.”
I’d like the authors to clarify whether this study was demonstrating systems described or if it also involved trialing the system on real people over a number of meals. I was not completely sure based on the way the study was written.
-> (ans.) In this paper, we authors have demonstrated a potential solution as a meal monitoring system. It is not yet involved to trial on real patients. However, it would be our future-goal to exploit this combined system on clinical beds for practical tests and eventually make products to supply for hospitals and nursing centers.
While I understand communication articles are shorter than research article formats, the authors should still provide a brief discussion section to address the strengths and weaknesses of the systems, as well as issues regarding their implementation in real world situations. E.g. you mentioned customizing trays with cell placement for weight monitoring, but does that mean LTCFs have to change all their trays? How would that influence costs?
-> (ans.) Thanks a lot for this comment. It would be a great burden for LTCFs to change all the trays in terms of “unexpected expense and staffs to use”. Therefore, in this work, we would fit and adjust this innovative device to a normal food-tray that is already utilized in LTCFs, hospitals, or nursing homes. Particularly, the application of low-cost Arduino boards would help to maximize the feasibility with cost reduction.

Round 2
Reviewer 1 Report
The authors have revised and/or responded accordingly.
Author Response
We authors do appreciate your valuable comments and suggestions that helped our manuscript improve a lot.
Best regards,
Sung Min Park